# Assessment of organizational readiness to implement an electronic health record system in a low-resource settings cancer hospital: A cross-sectional survey

Johnblack K. Kabukye[1,2]*, Nicolet de Keizer[2], Ronald Cornet[2]

**1** Uganda Cancer Institute, Kampala, Uganda, **2** Department of Medical Informatics, Amsterdam Public Health research institute, Amsterdam UMC—Location AMC, Amsterdam, The Netherlands

* jkabukye@gmail.com

## Abstract

### Background

Organizational readiness for change is a key factor in success or failure of electronic health record (EHR) system implementations. Readiness is a multifaceted and multilevel abstract construct encompassing individual and organizational aspects, which makes it difficult to assess. Available tools for assessing readiness need to be tested in different contexts.

### Objective

To identify and assess relevant variables that determine readiness to implement an EHR in oncology in a low-and-middle income setting.

### Methods

At the Uganda Cancer Institute (UCI), a 100-bed tertiary oncology center in Uganda, we conducted a cross-sectional survey using the Paré model. This model has 39 indicator variables (Likert-scale items) for measuring 9 latent variables that contribute to readiness. We analyzed data using partial least squares structural equation modeling (PLS-SEM). In addition, we collected comments that we analyzed by qualitative content analysis and sentiment analysis as a way of triangulating the Likert-scale survey responses.

### Results

One hundred and forty-six clinical and non-clinical staff completed the survey, and 116 responses were included in the model. The measurement model showed good indicator reliability, discriminant validity, and internal consistency. Path coefficients for 6 of the 9 latent variables (i.e. vision clarity, change appropriateness, change efficacy, presence of an effective champion, organizational flexibility, and collective self-efficacy) were statistically significant at $p < 0.05$. The $R^2$ for the outcome variable (organizational readiness) was 0.67. The sentiments were generally positive and correlated well with the survey scores (Pearson's r = 0.73). Perceived benefits of an EHR included improved quality, security and accessibility of

**Funding:** This study was made possible through a scholarship to JKK by the Uganda Cancer Institute, with funding support from the African Development Bank (AfDB), under the East Africa Regional Centre of Excellence in Oncology project (Project ID P-Z1-IB0-024) https://www.afdb.org/en/projects-and-operations/p-z1-ib0-024 The funders had no role in study design, data collection and analysis, decision to publish, or preparation of the manuscript.

**Competing interests:** The authors have declared that no competing interests exist.

**Abbreviations:** AVE, Average Variance Extracted; EHR, Electronic Health Record; LMIC, Low and middle income country; PLS, Partial Least Squares; SEM, Structural Equation Modeling; UCI, Uganda Cancer Institute.

clinical data, improved care coordination, reduction of errors, and time and cost saving. Recommended considerations for successful implementation include sensitization, training, resolution of organizational conflicts and computer infrastructure.

## Conclusion

Change management during EHR implementation in oncology in low- and middle- income setting should focus on attributes of the change and the change targets, including vision clarity, change appropriateness, change efficacy, presence of an effective champion, organizational flexibility, and collective self-efficacy. Particularly, issues of training, computer skills of staff, computer infrastructure, sensitization and strategic implementation need consideration.

## Introduction

Electronic health record (EHR) systems are postulated and have been demonstrated to improve healthcare safety, efficiency and overall quality through improved care coordination, reduction of medical errors, saving time and costs and enhancement of collection of quality healthcare data to support clinical research and healthcare management [1–3].

However, EHR implementation is a complex and challenging organizational change which is often resisted with planned or actual boycotts, and workarounds by medical staff to state-of-the-art systems [3,4].Although failures are not commonly reported in literature [3], it is estimated that 50–75% of implementations of EHRs and other health information technologies fail–i.e. they overrun budgets or implementation time, do not provide end user satisfaction, or are completely abandoned [3–7].Implementation of EHRs is difficult because it is not merely a technological change, but rather a socio-technical change process that affect many aspects of the organization [8–10]. It often results into changes or disruptions in clinical workflows, introduction of extra tasks, or shifting of tasks from one cadre to another [3–5,11,12] e.g. patients entering clinical history via patient portals or medical assistants and front desk refilling prescriptions, a task usually done by physician and pharmacists [13]. In addition, EHR implementation often requires learning of new (computer) skills or applications, and comes with actual or perceived changes in the power structure and legal responsibilities within healthcare, such as threat to doctors' autonomy when computerized clinical decision support functionality is implemented [3–7,12].

Organizational readiness for change is a well-known factor that influences success of organizational changes in general, and in EHR implementation in particular [3,7,12,14–20,21–26]. It is a multifaceted and multilevel construct, and therefore can be difficult to measure. Holt et al. [14]discuss four facets of readiness covering (i) the change process, i.e. the steps and strategies followed during implementation of the change, e.g., extent of stakeholder involvement, (ii) the content of the change, i.e. the particular initiative being implemented such as the EHR system and its characteristics, (iii) the context of the organization including the conditions and environment under which staff work, e.g., dynamic, learning organizational culture, financial and human resource capacity, and (iv) individual attributes of the staff or those affected by the change, e.g., their skills, biases and prejudices. Different forms of readiness have been described in literature with some overlap in meaning. Examples include core (need/motivational) readiness, technological (infrastructural) readiness, societal readiness, engagement

readiness and learning (IT skills) readiness [16–18,26].The relative importance of each of these forms of readiness varies between organizational contexts. For example, poor IT infrastructure and lack of IT skills are often a barrier for EHR implementation in LMICs making technological readiness relatively more important for LMICs [18,19,26–29].

Weiner [17] conceptualizes organizational readiness as the extent to which organization staff are psychologically and behaviorally prepared. That is, the extent to which they are willing (change commitment) and able (change efficacy) to make and maintain the change. Weiner's unified view of readiness at the organizational level is motivated by the premise that healthcare improvement interventions such as EHR implementation "entail collective behavior change in the form of systems redesign–that is, multiple, simultaneous changes in staffing, work flow, decision making, communication, and reward systems". According to Weiner, the above mentioned forms of readiness are antecedents to organizational readiness. Change commitment, and thus motivation to take the change action, comes when staff feel that they *want*–i.e. they value the change–as opposed to when they feel that they *have to*–i.e. when they feel they have no option and are obliged to take the action [17].For staff *to want* to make the change, they must be dissatisfied with the current state, and appreciate or be convinced about the advantage of the future state. Change efficacy (i.e. organization staff's belief in their capabilities to accomplish the change action or belief that successful change is possible, e.g., from stories of success from similar organizations) depends on staff's understanding and judgment of the task demands (what it takes to effect the change) and the available resources such as finances or IT infrastructure [17].

Kotter [15] argues that half of large organizational changes fail because of lack of readiness. Organization staff seek to maintain a state of affairs that provides them a sense of psychological safety, control and identity; and any attempts to change this status quo is resisted [14–17]. A process of "unfreezing" must occur in which mindsets are changed and motivation for change created [16]. Shea et al note that "when organizational readiness is high, members are more likely to initiate change, exert greater effort, exhibit greater persistence, and display more cooperative behavior, which overall results in more effective implementation of the proposed change" [20].

Early perceptions and beliefs about the change play a central role in shaping future attitudes and behaviors such as negative rumors, involvement in the planning and design phases, and resistance to change [24]. It is thus crucial to assess readiness prior to major organizational change such as EHR implementation in order to ensure higher chances of success [7,17,18,20,24]. Conducting a readiness assessment helps uncover action points or issues that threaten success and these can be addressed early in the project lifecycle when change management is most efficient [17,18,24]. Moreover, the readiness assessment process itself can increase the readiness as it introduces the impending change to the organization staff and spurs discussion.

Several tools have been published for measuring readiness both at organizational level as well as at individual level in different contexts. Kamisah and Yusof [21] have reviewed tools and models for measuring readiness in information system adoption and conclude that measuring readiness at the organizational level is more advantageous than at individual level, and also that there is no single best model or measure for all circumstances. Gagnon et al. [22] have conducted a systematic review of tools (models and questionnaires) for assessing readiness in healthcare where they found that many lacked information on reliability and validity, and needed to be tested in diverse clinical contexts.

In this study we aimed to determine which factors within the model by Paré et al. [24] underlie perceived organizational readiness to implement an EHR in oncology in Low and Middle Income Countries (LMICs). We chose the Paré model because it measures readiness at

organizational level compared to, for example, tools by Khoja et al [18] which measure readiness at the level of antecedent constructs [17]. Moreover, the Paré model was developed and validated within the same context of cancer care as the Uganda Cancer Institute, where this study was conducted.

## Materials and methods

### Study design

We conducted a cross-sectional survey based on the model and questionnaire developed by Paré et al. [24] which is open access under a Creative Commons (CC-BY) license. As shown in **Fig 1**, the Paré model consists of ten latent constructs or variables: Vision clarity, Change appropriateness, Change efficacy, Top-management support, Presence of an effective champion, Organizational history of change, Organizational politics and conflicts, Organizational flexibility, Collective self-efficacy and Organizational readiness. In the model, Organizational readiness is referred as an endogenous latent variable because it is essentially an outcome variable which the other nine (referred to as exogenous latent variables) measure. The nine exogenous variables fall under 4 facets similar to those discussed by Holt [14].

All latent variables are measured on four Likert-scale questionnaire items (referred to as manifest or indicator variables), except Presence of a champion which is measured on three items. This makes a total of 39 questionnaire items. In our study, the scale was 5-point, with 5 = strongly agree, 4 = agree, 3 = neither agree nor disagree, 2 = disagree, 1 = strongly disagree.

We also added sections for comments to encourage participants to give more details to explain why they scored the organization the way they did. Respondent characteristics including age, gender, tenure, computer usages, and prior EHR experience were also collected since these affect readiness [18,23]. **S1 File** shows the questionnaire.

### Setting

The study was conducted at the Uganda Cancer Institute (UCI) [30], a 100-bed tertiary public cancer hospital in Kampala, Uganda. The UCI receives about 5000 new cancer patients per year from Uganda and neighboring countries. Clinical documentation is done on paper. However, three years ago the UCI procured an off-the-shelf EHR called Clinic Master which currently is only being used for patient registration, appointments scheduling, and retrospective capture of some clinical details such as diagnosis and treatment, as well as for tracking paper files. Only a few of the staff directly interact with the EHR, mostly the biostatisticians and data entry clerks. The system has provisions for capturing free-text clinical notes, as well as billing, ordering of lab investigations, etc., but these functionalities are not yet being used. Efforts are ongoing to customize Clinic Master to suit the exact needs of the users with regards to cancer care workflow, as well as considerations to switch to a different system altogether.

### Participants selection and questionnaire distribution

Eligible participants were all UCI staff who are directly involved in patient care or directly use the EHR. There are approximately 250 of these staff. Printed questionnaires were used for data collection, and they were distributed by the first author who is one of the clinical staff at the UCI. Additionally, staff who were not on site during the survey period (September to October 2018), e.g. due to study leave or other travels, were excluded since paper questionnaire were used. Online questionnaires were discouraged by the research ethics committee because there was no way to stamp them and assure the participants of official approval.

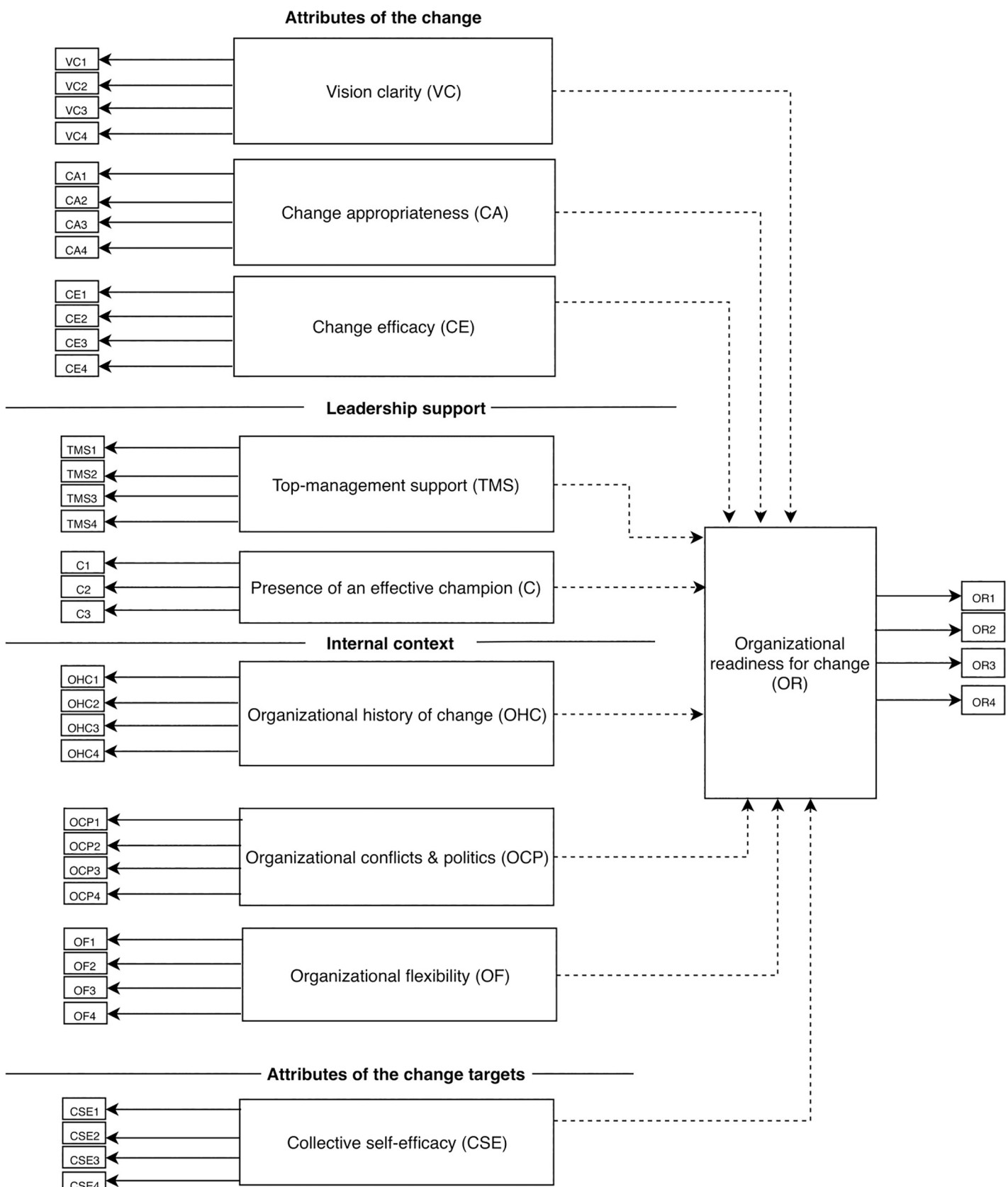

**Fig 1. Research model.** Solid arrows show paths from indicator variables (questionnaire item) to the latent variables. Dashed arrows show paths from exogenous latent variable to the endogenous latent variable. See Paré et al. [24] for definitions of the constructs.

To minimize ordering effect [31], we made five versions of the questionnaire containing exactly the same items but with their order randomly shuffled using the free online list randomizer [32]. The questionnaires were in English. One hundred and seventy-five questionnaires were distributed and 146 were returned (83.4% response rate).

Using G\*Power [33] v3.1.9.2, the calculated minimum sample size required to detect a small effect size ($R^2$ of 0.3) in our model where the maximum number of predictors (or arrows pointing at a latent variable) is 9, at a significance level 5% and statistical power of 80%, is 62 cases. Alternatively, following the rule of thumb [34], the minimum sample size for our model is 90 –i.e. 10 times the maximum number of predictors.

## Data analysis

Double data entry was done using Epi Data v4.4.2.1 [35]by two independent data clerks and any transcription errors were resolved. We performed descriptive statistics using SPSS v24 [36].

For model analysis, we used the R statistical environment [37], specifically theplspm package v0.4.9 [38], to perform structural equation modeling (SEM) using the partial least squares (PLS) method. We reverse-coded negatively phrased indicator variables to correct their direction with respect to the latent construct, and removed all cases with missing values in any of the 39 indicator variables (questionnaire items, **S1 File**) since the PLS algorithm requires complete cases. Details of SEM and PLSare provided in **S2 File**, and the Data and R code in **S4 File**.

We tested our model using measures as described in Hair et al. [34]. **Table 1**shows the measures for validating the measurement model, i.e. loadings or communalities for indicator reliability, cross loadings for discriminant reliability, Dillon-Goldstein's rho for composite reliability, and average variance extracted (AVE) for convergent validity.

We tested the structural model using the $R^2$ (also called the coefficient of determination) for the endogenous latent variables, as well as the path coefficients for the exogenous latent variables. The $R^2$ indicates the amount of variance in the endogenous latent variable that is explained by the exogenous latent variables. $R^2$ values <0.3 are considered low, between 0.3 and 0.6 moderate, and above 0.6 are high [39].

We also conducted sentiment analysis of the comments from the survey using the R package sentiment [40], to determine the overall polarity i.e. how negative or positive respondents felt about the UCI's readiness for change.

**Table 1. Performance measures for model validation as described by Hair et al [34].**

| Measure | Type of validation | Target |
|---------|-------------------|--------|
| Outer loadings (or the squared outer loadings called communalities) | Indicator reliability–i.e. if each indicator has significant contribution to measuring the respective latent variable | > 0.708 (or communality of > 0.5) |
| Cross loadings | Discriminant validity–i.e. how the indicator variable loads on its respective latent variable vs on other latent variables | outer loadings are higher for the respective latent variable compared to other latent variables (meaning the indicator variable measures the latent variable it is supposed to measure and not other latent variables) |
| Dillon-Goldstein's rho (similar to Cronbach's alpha but allows the indicator variables to have varying outer loadings) | Composite reliability (internal consistency)–i.e. if the indicator variables are correlated, (meaning they measure the same latent variable) and in the same direction | > 0.7 (or >0.6 in exploratory research) |
| Average variance extracted (AVE) | Convergent validity–measures how much of the variance of the indicator variable is captured by the latent variable in relation to measurement error | > 0.5 |

As a way of triangulation, we used the mean score of each indicator variable and the sentiment score of the corresponding comment to calculate as correlation (Pearson's *r*).Similar to model analysis, we also reverse-coded negatively phrased indicator variables for correlation analysis.

Lastly, we conducted deductive content analysis of the comments [41]using the R package RQDA [42] to derive perceived benefits or reasons to implement the EHR as well as action points to get the organization ready.

### Ethics and consent to participate

The study was reviewed and approved by the UCI Research Ethics Committee #UCIREC 12–2018, and by Uganda National Council of Science and Technology #IS14ES. Each participant was given an informed consent form to read and sign before filling in the questionnaire.

## Results

### Respondents

One hundred and forty-six respondents completed the questionnaire, which is 58.4% of the target population and 83.4% response rate. **Table 2** shows the participant characteristics.

About 72% were 40 years or younger, 59% were female, and 75% had worked at the organization for 1–10 years. Eighty-three percent of respondents were clinical (oncologists, general doctors, nurses and allied health workers), 89% reported using computers at least on a weekly basis, with 80% rating their computer skills as intermediate to advanced. Fifty-six percent reported experience using an EHR, but only 40.4% reported ever receiving EHR training.

### Model analysis

Thirty cases (20%) had missing values in at least one of the indicator variables needed for model analysis, so they were removed from the analysis, leaving 116 cases. The pattern of missing values was random.

Twenty-five of the 39 indicator variables had loadings above the cutoff of 0.708 which implies good indicator reliability. The loadings are shown in **Table 3** along with communalities and weights. Only change appropriateness, presence of an effective champion, and top-management support, had all indicator variables with loadings above the cutoff. The remaining six latent variables had at least one indicator variable with a loading above the cutoff.

The cross loadings show good discriminant validity as evidenced by all indicator variables loading highest on their respective latent variables (see **S3 File**)

All latent variables had good internal consistency as indicated by Dillon-Goldstein's rho of 0.7 or higher, although Organizational history of change was borderline (**Table 4**).Vision clarity, change appropriateness, top-management support, presence of a champion, and collective self-efficacy, showed good convergent validity, i.e. AVE above the cut-off value of 0.5.For organizational readiness, the only endogenous latent variable in the model, $R^2 = 0.67$.

Path coefficients for vision clarity, change appropriateness, change efficacy, presence of an effective champion, organizational flexibility, and collective self-efficacy, were statistically significant at $p < 0.05$ (**Table 4**).

### Qualitative analysis

Results for sentiment analysis of the comments on each of the items (indicator variable) and one general comment are shown in **Table 3**, along with the mean and standard deviation for each indicator variable. The sentiment scores ranged from -0.113 to +0.4, but generally were

**Table 2. Characteristics of respondents.**

| | | n | % |
|---|---|---|---|
| Total | | 146 | 100 |
| **Gender** | | | |
| | Female | 86 | 58.9 |
| | Male | 53 | 36.3 |
| | Missing | 7 | 4.8 |
| **Age ranges** | | | |
| | 30 yrs or younger | 47 | 32.2 |
| | 31–40 | 58 | 39.7 |
| | 41–50 | 20 | 13.7 |
| | 50 yrs or older | 13 | 8.9 |
| | Missing | 8 | 5.5 |
| **Tenure (How long have you been working in this organization?)** | | | |
| | 1 yr or less | 19 | 13.0 |
| | >1 yr—5 yrs | 53 | 36.3 |
| | >5 yrs—10 yrs | 57 | 39.0 |
| | > 10yr | 11 | 7.5 |
| | Missing | 6 | 4.1 |
| **Job title** | | | |
| | Oncologist (consultants) | 9 | 6.2 |
| | Doctor | 27 | 18.5 |
| | Nurse | 24 | 16.4 |
| | Allied health worker (lab, imaging, pharmacy, medical records officers) | 61 | 41.8 |
| | Biostatistics/Data manager/IT | 13 | 8.9 |
| | Administrator | 12 | 8.2 |
| **Frequency of computer usage** | | | |
| | Daily | 101 | 69.2 |
| | A few times a week | 29 | 19.9 |
| | A few times a month | 11 | 7.5 |
| | A few times a year | 1 | 0.7 |
| | Never | 4 | 2.7 |
| **Computer proficiency (self-assessment): 1 = Basic computer skills (need help with internet and email or office applications), 5 = Proficient (able to do advanced tasks such as database management or programming)** | | | |
| | 1 (Basic) | 17 | 11.6 |
| | 2 | 12 | 8.2 |
| | 3 | 47 | 32.2 |
| | 4 | 44 | 30.1 |
| | 5 (Advanced) | 26 | 17.8 |
| **Experience using electronic health record systems (EHR)** | | | |
| | Yes | 82 | 56.2 |
| | No | 64 | 43.8 |
| **Ever received training on electronic health record systems (EHR)** | | | |
| | Yes | 59 | 40.4 |
| | No | 87 | 59.6 |

positive. Comments for TMS3 (Top-management support), OCP2, OCP4 (Organizational conflicts and politics), OF4 (Organizational flexibility) and CSE1 and CSE3 (Collective self-efficacy) had negative sentiment; while the rest had positive sentiment. The general comment

**Table 3. Mean scores, standard deviations, loadings, communalities and weights for the indicator variables (Questionnaire items) across all respondents, and sentiment scores for comments made against each item.** Item scores are: 1 = Strongly disagree, 2 = Disagree, 3 = Neither agree nor disagree, 4 = Agree, 5 = Strongly agree. Items in *italics* were reverse-code before mean and SD was calculated to correct their direction with respect to the latent construct. Mean score of 3 ("neither agree nor disagree") or above, Loadings and Communalities above the cutoff, and Positive sentiments are **bolded**. Variables adapted from [24].

| Latent variable | Questionnaire item | Indicator variable | Mean | SD | Loading | Communality | Weight | Sentiment |
|---|---|---|---|---|---|---|---|---|
| Vision clarity (VC) | I believe there are legitimate reasons for us to introduce a computer-based system in our unit. | VC1 | **4.55** | 0.69 | 0.69 | 0.48 | 0.29 | **0.40** |
| | We definitely need new tools to improve the way we work around here. | VC2 | **4.61** | 0.74 | **0.81** | **0.66** | 0.34 | **0.21** |
| | There are a number of rational reasons for the deployment of EHR system in our unit. | VC3 | **4.32** | 0.81 | 0.70 | 0.48 | 0.29 | **0.27** |
| | A computer-based system is needed to improve our clinical processes. | VC4 | **4.53** | 0.76 | **0.83** | **0.69** | 0.39 | **0.29** |
| Change appropriateness (CA) | I think that staff in our unit will benefit from the use of an EHR. | CA1 | **4.51** | 0.79 | **0.84** | **0.71** | 0.32 | **0.26** |
| | The deployment of an EHR will contribute to our unit's overall performance. | CA2 | **4.49** | 0.78 | **0.79** | **0.62** | 0.27 | **0.23** |
| | The deployment of an EHR matches the priorities of our unit. | CA3 | **4.06** | 1.04 | **0.71** | **0.51** | 0.29 | **0.09** |
| | The implementation of an EHR will prove to be best for our unit. | CA4 | **4.47** | 0.78 | **0.86** | **0.74** | 0.35 | **0.28** |
| Change efficacy (CE) | I know staff outside our unit who had successful experiences with an EHR. | CE1 | **3.81** | 1.16 | 0.60 | 0.36 | 0.29 | **0.02** |
| | An EHR has been successfully deployed in clinical units similar to ours. | CE2 | **3.54** | 1.21 | 0.58 | 0.33 | 0.31 | **0.10** |
| | An EHR has received positive reviews in the press (e.g., newspapers, magazines, seminars, etc) | CE3 | **3.67** | 1.00 | 0.63 | 0.40 | 0.25 | **0.12** |
| | I believe the government/ministry's movement toward the electronic medical record represents a driving force for the deployment of an EHR in our unit | CE4 | **3.88** | 0.99 | **0.82** | **0.67** | 0.60 | 0.00 |
| Top-management support (TMS) | Managers in our unit are committed to the deployment of an EHR. | TMS1 | **3.60** | 1.01 | **0.78** | **0.60** | 0.34 | **0.19** |
| | Managers in our unit have stressed the importance of this change. | TMS2 | **3.36** | 1.11 | **0.76** | **0.58** | 0.36 | **0.12** |
| | Managers have sent a clear message that the deployment of an EHR will occur in our unit. | TMS3 | **3.13** | 1.10 | **0.73** | **0.54** | 0.28 | -0.11 |
| | Staff have been encouraged to embrace the upcoming deployment of an EHR. | TMS4 | **3.63** | 1.04 | **0.79** | **0.62** | 0.33 | **0.17** |
| Presence of an effective champion (C) | There is a champion who actively promotes the deployment of an EHR in our unit. | C1 | **3.67** | 1.11 | **0.85** | **0.73** | 0.48 | **0.24** |
| | The EHR project has a credible and trustworthy champion. | C2 | **3.95** | 0.84 | **0.80** | **0.63** | 0.43 | **0.12** |
| | There is a champion who will be able to push the EHR project over or around implementation hurdles. | C3 | **3.92** | 0.95 | **0.72** | **0.52** | 0.34 | **0.00** |
| Organizational history of change (OHC) | Our unit has successfully implemented other technological changes in recent years. | OHC1 | **3.23** | 1.11 | 0.59 | 0.35 | 0.26 | **0.13** |
| | *Staff in our unit have had negative experiences with technological projects in the past.* | OHC2 | **3.15** | 1.16 | 0.19 | 0.04 | 0.15 | **0.01** |
| | Our unit is usually successful when it undertakes all types of changes. | OHC3 | **3.48** | 1.03 | **0.73** | **0.53** | 0.49 | **0.22** |
| | Information technology initiatives have been encouraged and are common practices in our unit. | OHC4 | **3.45** | 1.04 | **0.84** | **0.70** | 0.55 | **0.04** |

(*Continued*)

**Table 3.** (Continued)

| Latent variable | Questionnaire item | Indicator variable | Mean | SD | Loading | Communality | Weight | Sentiment |
|---|---|---|---|---|---|---|---|---|
| Organizational conflicts and politics (OCP) | Mutual trust and cooperation among staff in our unit is strong. | OCP1 | **3.50** | 1.02 | **0.86** | **0.74** | 0.73 | **0.06** |
| | *Recent attempts to change the way we work in our unit have been hindered by political forces or conditions.* | OCP2 | 2.80 | 1.29 | 0.58 | 0.33 | 0.34 | -0.11 |
| | *The climate in our unit is mainly characterized by conflicts and disputes.* | OCP3 | **3.51** | 1.16 | 0.57 | 0.33 | 0.23 | **0.01** |
| | *Staff frustration is common in our unit.* | OCP4 | 2.72 | 1.23 | 0.43 | 0.19 | 0.11 | -0.10 |
| Organizational flexibility (OF) | Our unit is structured to allow superiors to make changes quickly. | OF1 | **3.30** | 1.11 | **0.74** | **0.54** | 0.45 | **0.02** |
| | It is easy to change procedures in our unit to meet new conditions. | OF2 | **3.46** | 1.08 | **0.75** | **0.57** | 0.41 | **0.13** |
| | *Getting anything changed in our unit is a long, time-consuming process.* | OF3 | 2.85 | 1.29 | 0.47 | 0.22 | 0.20 | **0.05** |
| | Policies and procedures in our unit allow us to take on new challenges effectively | OF4 | **3.50** | 1.08 | **0.75** | **0.56** | 0.35 | -0.03 |
| Collective self-efficacy (CSE) | All staff in our unit are highly computer literate. | CSE1 | 2.80 | 1.16 | **0.79** | **0.63** | 0.35 | -0.02 |
| | It won't take a long time before staff in our unit feel comfortable using an EHR. | CSE2 | **3.54** | 1.18 | **0.78** | **0.61** | 0.42 | **0.15** |
| | Using a computer effectively is no problem for the staff in our unit. | CSE3 | **3.22** | 1.22 | **0.78** | **0.62** | 0.32 | -0.01 |
| | *In general, staff in our unit have low computer skills.* | CSE4 | **3.02** | 1.19 | 0.66 | 0.44 | 0.23 | **0.09** |
| Organizational readiness (OR) | I believe an EHR can be successfully implemented in our unit. | OR1 | **4.38** | 0.73 | **0.73** | **0.53** | 0.40 | **0.14** |
| | *Managers should delay the deployment of an EHR in our unit.* | OR2 | **4.40** | 0.84 | 0.53 | 0.28 | 0.20 | **0.21** |
| | The deployment of an EHR in our unit is timely. | OR3 | **3.99** | 1.06 | 0.63 | 0.40 | 0.29 | **0.01** |
| | Our unit is ready to take on this technological change. | OR4 | **3.95** | 1.02 | **0.84** | **0.70** | 0.50 | **0.11** |
| | General comment | | | | | | | **0.21** |

had a sentiment score of +0.23. The sentiment scores for the comments were strongly correlated with the mean scores of the corresponding indicator variable, Pearson's $r = 0.73$.

Table 5 shows reasons for implementing an EHR and action points/considerations according to content analysis of the comments from the respondents. The respondents consider the

**Table 4. Results of performance measures for model validation.** Values in **bold** are significant or above threshold. Cronbach's alpha is provided for comparison with Paré et al [24].

| Latent variable | Latent variable type | # of indicator variables | Dillon-Goldstein's rho | Cronbach's alpha | R2 | AVE | Path coefficients | P value |
|---|---|---|---|---|---|---|---|---|
| Vision clarity (VC) | Exogenous | 4 | **0.84** | **0.75** | - | **0.58** | 0.16 | **0.0239** |
| Change appropriateness (CA) | Exogenous | 4 | **0.88** | **0.82** | - | **0.65** | 0.24 | **0.0050** |
| Change efficacy (CE) | Exogenous | 4 | **0.77** | 0.60 | - | 0.44 | 0.17 | **0.0140** |
| Top-management support (TMS) | Exogenous | 4 | **0.85** | **0.77** | - | **0.59** | -0.02 | 0.8264 |
| Presence of an effective champion (C) | Exogenous | 3 | **0.84** | 0.70 | - | **0.63** | 0.15 | **0.0299** |
| Organizational history of change (OHC) | Exogenous | 4 | **0.70** | 0.46 | - | 0.40 | 0.07 | 0.3729 |
| Organizational conflicts and politics (OCP) | Exogenous | 4 | **0.77** | 0.59 | - | 0.40 | 0.02 | 0.7071 |
| Organizational flexibility (OF) | Exogenous | 4 | **0.78** | 0.63 | - | 0.47 | 0.22 | **0.0037** |
| Collective self-efficacy (CSE) | Exogenous | 4 | **0.85** | **0.76** | - | **0.57** | 0.21 | **0.0045** |
| Organizational readiness (OR) | Endogenous | 4 | **0.79** | 0.64 | **0.67** | 0.48 | - | - |

**Table 5. Reasons for implementing an EHR and Action points/Key considerations.**

| Code | Freq | Sample quote |
|---|---|---|
| **Why EHR (Perceived benefit)** | | |
| Improve data quality, security and accessibility | 44 | "if the databases are well managed evidence based solution are quick to find because the data is readily available" |
| | | "timely reporting, monitoring patients outcomes and just a click away for data sharing, analysis and interpretation" |
| | | "paper work gets lost ad makes the place untidy but when you use soft copy patients information will be kept safe" |
| Improve coordination, communication and consultation | 22 | "EHR will improve inter departmental communication which reduces patient review time" |
| Save time | 20 | "It will shorten the turnaround time for example receiving lab results, images as sometimes there are delays in picking" |
| "because everyone is doing it" | 16 | "As technology advances we definitely need to move with the tide" |
| | | "EHR is strongly recommended and encouraged in many facilities; in fact most private facilities have implemented it" |
| Improve accountability and stock management | 6 | "I have seen different hospitals greatly manage their stock using this system. This is a big institution with many patients; this move will ease work in my unit through controlling the way drugs move in and out of our unit, knowing the previous diagnosis and drugs issued out" |
| Save money/resources | 3 | "There has been long term use of paper records. With limited resources for recording materials [and] increasing number of clients, this makes me feel the organization is ready to adapt to EHR" |
| Reduce errors | 2 | "..since each medical personnel will easily access the patient's information, errors will also be minimized" |
| **Action points/ Key considerations** | | |
| Training—initial and ongoing | 30 | "In order for the EHR system to be successful staff need to be trained and familiarised with the [system]" |
| Advocacy and sensitization, particularly seniors or managers | 28 | "some senior staff who would support the implementation of the EHR change still have negative attitude towards the need for change. Also, I think people have fear that they may lose their jobs if they implement EHR" |
| Lack of computer skills | 16 | "Some staffs have low computer skills so using a computer effectively is not easy" |
| Under-staffing | 14 | "But before introducing it on the ward let them first think of staff because we cannot be 2 nurses on day duty 1 nurse on evening and night shift and you think I will be in position to enter the information in the computer" |
| Strategic implementation process | 12 | "It will require a careful, coordinated roll out . . . over months to years. . ." |
| | | "Let our leaders in the department be involved when some of this technology is being planned for" |
| IT infrastructure | 12 | "EHR needs a lot of (infra) structural support–reliable power, trustworthy backups and trust of data safety in the IT" |
| | | "In our unit we have only one computer" |
| Organizational conflicts and inertia | 7 | "There is a lot of ground politics and sticking on policies. Negative attitude of groups or individuals about new technology, at times people have to be dragged into it to appreciate changes" |
| | | "There is conflict of top managers which hinders the use of EHR– some say use paper work and others electronic" |
| Funding | 4 | "I think our organization is not yet ready to implement EHR due to financial constraints" |
| Other competing priorities | 3 | "I think there are more basic issues to be addressed first e.g. timely investigation results, chemotherapy and antibiotic availability, blood products and stationery" |

(*Continued*)

**Table 5.** (Continued)

| Code | Freq | Sample quote |
|---|---|---|
| Space for computers | 3 | "Space for IT systems is lacking in the clinical areas" |
| Government policies | 1 | "However, due to government policies there may be some delays in implementing things which would be of use to organizations" |

EHR important for improving clinical data quality, security and accessibility; for improvement of communication and care coordination, save time, reduce errors, improve accountability and stock management, while others say an EHR should be implemented because it is the trend and other hospitals are implementing one. Key considerations for successful implementation suggested by the respondents include advocacy and sensitization about the change, training of staff on computer skills and specifically on the EHR, and addressing the issues of understaffing and inadequate computer infrastructure needed for implementing the EHR.

## Discussion

In this study we assessed the variables within the Paré model [24] that contribute to organizational readiness for change in the context of EHR implementation in oncology in LMICs. We also gained insights on the level of readiness of the study organization, the UCI, to implement an EHR. The Paré model, originally developed and tested within the context of oncology and mental health in Canada, consists of 9 theory-based variables associated with organizational readiness for change. These relate to the attributes of the change, attributes of the change targets, leadership support and internal context of the organization.

Besides our study being done in an LMIC, our participants were more diverse in terms of profession compared to Paré et al., which largely involved nurses. Additionally our participants were relatively younger and had had shorter tenures.

Despite the above differences, our results were similar to those of Paré et al. in showing that, based on the $R^2$, generally the model performs well in measuring organizational readiness, and the questionnaire has good validity and reliability as shown by the loadings and cross loadings, Dillon-Goldstein's rho and AVE.

However, 14 of the 39 indicator variables showed poor reliability. Specifically, 3 of the 4 indicator variables in change efficacy (i.e. CE1, CE2, and CE3) and in organizational conflicts and politics (i.e. OCP2, OCP3 and OCP4), had low loadings. If indicator variables with low reliability are eliminated from the final model and questionnaire, these latent variables will remain with only one indicator variable, and the whole questionnaire will have 25 items instead of 39 which makes it shorter. Paré's final questionnaire had 35 items after eliminating OHC2, OHC4 (Organizational history of change), OCP2, and OF4 (Organizational flexibility).

The change efficacy variable concerns staff being inspired by EHR implementation projects from other organizations similar to theirs. The low reliability for change efficacy in our study is likely due to the fact that the study site, the UCI, is the only oncology center in the country, and therefore respondents did not have similar hospitals to compare with or get inspiration. Organizational conflict and politics had low reliability yet from the qualitative findings (comments). This was a frequently mentioned point to consider. This is likely due to the high-context culture of Uganda [43]–i.e. people prefer to avoid conflict, do not give direct feedback and hesitate to discuss issues around organizational conflicts, staff frustration, corporation and trust even when these issues are a reality.

In addition, only 6 of the theorized 9 latent variables are supported by our findings as significantly contributing to measurement of organizational readiness based on p-values <0.05.

These are: vision clarity, change appropriateness, change efficacy, presence of an effective champion, organizational flexibility, and collective self-efficacy, which fall under attributes of the change and attributes of the change targets.

These findings suggest that change management during EHR implementation at this organization, and others similar to it, should focus on making sure that all staff understand why the EHR being implemented (vision clarity), convincing the staff that the EHR is appropriate and will improve their work (change appropriateness), and ensuring that staff, individually and collectively, have the required skills, motivation, inspiration and resources for successful EHR implementation (change efficacy and collective self-efficacy). It is also important that there is an influential and respected person to champion the implementation process. Champions are important for success of EHR implementation because, as early adopters with positive attitude and enthusiasm towards the impending change, they help in communicating the expected benefits to their peers and encourage them to adopt the change [44–47]. Organizational flexibility, which is also significant, might not be very actionable since it is historical, but measures can be put in place to improve it, for example, having smaller units within the organization which might accelerate change processes compared to rolling out an EHR in the entire organization.

The above impression is also supported by our qualitative findings, in that many of the action points/considerations relate to issues of sufficient staff, computer infrastructure, training and computer skills by the staff (efficacy), advocacy and sensitization of staff about the change (vision clarity), and careful execution of the change (change appropriateness).

The qualitative findings also show that the UCI is ready to implement an EHR, considering the fact that the staff understand what it will take to effect the change, and appreciate its benefits. The generally positive sentiments triangulate this conclusion.

The findings from this study have practical importance to both the study organization and other organizations. The UCI is in the process of EHR implementation albeit slow and with challenges. Whereas the decision whether to implement an EHR or not may not be solely based on the readiness assessment, findings from this study can give reassurance to the managers and EHR implementation team that the organization staff are ready for the EHR, and the action points or considerations suggested by the staff will help managers and project leaders to decide where to focus their efforts. Organizations similar to the UCI could use our findings to inform their own organizational change processes, either focusing on the variables in the model and the action points that we considered crucial for the UCI, or by testing the model in their own organizations to further confirm generalizability, as well as the predictability of implementation success by organizational readiness.

### Strength of the study

Collecting qualitative data provided a means to triangulate the quantitative data in the model, as well as giving it actionable meaning.

### Weakness of the study

The large proportion of missing values meant that about 20% of the responses were eliminated from model analysis. However, missing values were random, and the size of the remaining set was still larger than required according to sample size estimations. Another limitation is the use of data from one oncology center which might undermine generalizability.

### Conclusion

In this study we identified variables that are relevant for measurement of organizational readiness to implement an EHR in an oncology center in a low-income setting. These are: vision

clarity, change appropriateness, change efficacy, presence of an effective champion, organizational flexibility, and collective self-efficacy. In addition, we assessed organizational readiness and identified action points and considerations for enhancing readiness at a specific institution, UCI. We found that the UCI, while ready to implement an EHR, should pay attention to staff's computer skills, training of staff on EHR, available computer infrastructure, and should devise a strategic implementation plan. Whereas staff have a good understanding of the benefits of EHR implementation, which is important for high readiness, sensitization is also needed since some staff want to implement the EHR "just because everyone else is doing it".

## Supporting information

**S1 File. Questionnaire.**
(DOCX)

**S2 File. A brief explanation of SEM and PLS.**
(DOCX)

**S3 File. Cross-loadings.**
(DOCX)

**S4 File. Dataset and R code.**
(RAR)

## Acknowledgments

We thank Uganda Cancer Institute staff for filling in the survey. We also greatly appreciate Martine R. Groen, PhD, for her assistance with data analysis.

## Author Contributions

**Conceptualization:** Johnblack K. Kabukye, Nicolet de Keizer, Ronald Cornet.

**Data curation:** Johnblack K. Kabukye.

**Formal analysis:** Johnblack K. Kabukye, Nicolet de Keizer, Ronald Cornet.

**Funding acquisition:** Johnblack K. Kabukye.

**Investigation:** Johnblack K. Kabukye.

**Methodology:** Johnblack K. Kabukye, Nicolet de Keizer, Ronald Cornet.

**Project administration:** Johnblack K. Kabukye.

**Supervision:** Nicolet de Keizer, Ronald Cornet.

**Writing – original draft:** Johnblack K. Kabukye, Nicolet de Keizer, Ronald Cornet.

**Writing – review & editing:** Johnblack K. Kabukye, Nicolet de Keizer, Ronald Cornet.

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
