## [Decision Letter · Decision Letter 0]

22 Apr 2020

PONE-D-20-07679

Assessment of organizational readiness to implement an electronic health record system in a low-resource settings cancer hospital: structural equation modeling of survey data to identify relevant factors

PLOS ONE

Dear Dr. Kabukye,

Thank you for submitting your manuscript to PLOS ONE. After careful consideration, we feel that it has merit but does not fully meet PLOS ONE’s publication criteria as it currently stands. Therefore, we invite you to submit a revised version of the manuscript that addresses the points raised during the review process.

We would appreciate receiving your revised manuscript by Jun 06 2020 11:59PM. To enhance the reproducibility of your results, we recommend that if applicable you deposit your laboratory protocols in protocols.io, where a protocol can be assigned its own identifier (DOI) such that it can be cited independently in the future. For instructions see: http://journals.plos.org/plosone/s/submission-guidelines#loc-laboratory-protocols

We look forward to receiving your revised manuscript.

Kind regards,

Tim Luckett

Academic Editor

PLOS ONE

https://implementationscience.biomedcentral.com/articles/10.1186/1748-5908-9-7

https://www.ncbi.nlm.nih.gov/pmc/articles/PMC3056827/

The text that needs to be addressed is in the Introduction and Table 2.

In your revision ensure you cite all your sources (including your own works), and quote or rephrase any duplicated text outside the methods section. Further consideration is dependent on these concerns being addressed.

3. Please clarify in your Methods section whether the survey is published under a CC-BY license, or whether you obtained permission from the publisher to reproduce the questionnaire in this manuscript. Please explain any copyright or restrictions on this questionnaire.

Reviewers' comments:

Reviewer's Responses to Questions

**Comments to the Author**

1. Is the manuscript technically sound, and do the data support the conclusions?

Reviewer #1: Yes

Reviewer #2: Yes

2. Has the statistical analysis been performed appropriately and rigorously? 

Reviewer #1: Yes

Reviewer #2: Yes

3. Have the authors made all data underlying the findings in their manuscript fully available?

Reviewer #1: Yes

Reviewer #2: No

4. Is the manuscript presented in an intelligible fashion and written in standard English?

Reviewer #1: Yes

Reviewer #2: Yes

5. Review Comments to the Author

Reviewer #1: The study identifies and assesses relevant variables that determine readiness to implement an Electronic Health Record in oncology in a low- and middle-income setting by conducting a cross-sectional survey.

The study has importance, highlights perceived benefits of EHR as well as factors that support the successful implementation of EHR in Uganda. The study is a great addition to the emerging literature on readiness to implement health related interventions in low- and middle-income countries. More specific suggestions for improving the paper include:

1. The authors provided a succinct introduction that indicates the challenges related to the implementation of EHR. Majority of the references (except reference number 7 that included a study from Ethiopia in their review) the authors provided to support the EHR implementation barriers are from high-income countries (Page 4; line number 64-69). The authors missed the opportunity to differentiate between barriers that are predominant in LMICs and HICs when implementing EHR. However, if the barriers are uniquely similar, I suggest they make it clear.

2. The authors tried to conceptualise readiness in EHR; nevertheless, the introduction section lacks the necessary motivation and reflection explaining readiness in relation to low- and middle-income setting.

3. Applying the Paré model in Uganda is important to expand knowledge. However, the authors did not provide a strong rationale for adopting it (page 6; line number 114-117). Instead, the authors dedicated time explaining the Weiner’s theory of organisational readiness for change. What was the motivation for selecting the Paré model? What makes the Paré model relevant to apply within Uganda?

4. Th authors used “factors” in the title. Then, they used “variables” in the abstract (page 2; line number 24). Later, they returned to the term “factors” in the introduction (page 6; line number 114). Please, can the authors be consistent in the use of terminology?

5. The authors indicated that the questionnaires were printed and distributed (page 7; line number 144). However, it is not clear how participants were approached. Kindly provide details.

6. Authors indicated that not all UCI staff were on-site during the survey (page 8; line number 162-164). Please can the authors explain why they were unable to use an alternative approach such as an online-based or email-based survey to recruit the other staff?

7. Regarding the inclusion criteria, staff years of EHR experience was not considered. Why?

8. What was the relationship between the researcher who distributed the questionnaires and the participants prior to the study commencement? How did such a relationship influence the feedback? What strategies were integrated into the study to prevent/reduce potential bias from such researcher-participant relationship?

9. “…an influential and respected person to champion the implementation process” (Page 28; line number 314-315). Can the authors comment on and discuss with other international readiness literature the role of champions?

Reviewer #2: 1.The referred Fig. 1 is missing from the documents.

2. The tables can be re-organized to make them reader friendly.

3. Also, a number of them can be combined to reduce the number of tables

4. Other minor comments are as tracked in the word version of the manuscript attached

6. PLOS authors have the option to publish the peer review history of their article (what does this mean?). If published, this will include your full peer review and any attached files.

Reviewer #1: Yes: Andrew Donkor

Reviewer #2: Yes: Olutosin Awolude

---

## [Author Response · Author response to Decision Letter 0]

5 May 2020

Review Comments to the Author

Reviewer #1

The study identifies and assesses relevant variables that determine readiness to implement an Electronic Health Record in oncology in a low- and middle-income setting by conducting a cross-sectional survey.The study has importance, highlights perceived benefits of EHR as well as factors that support the successful implementation of EHR in Uganda. The study is a great addition to the emerging literature on readiness to implement health related interventions in low- and middle-income countries. More specific suggestions for improving the paper include

1. The authors provided a succinct introduction that indicates the challenges related to the implementation of EHR. Majority of the references (except reference number 7 that included a study from Ethiopia in their review) the authors provided to support the EHR implementation barriers are from high-income countries (Page 4; line number 64-69). The authors missed the opportunity to differentiate between barriers that are predominant in LMICs and HICs when implementing EHR. However, if the barriers are uniquely similar, I suggest they make it clear.

Response: Thank you for this suggestion. We have rewritten the introduction and included available literature summarizing barriers to EHR adoption in LMICs (Odekunle, Odekunle, and Shankar 2017; Saleh et al. 2016; Biruk et al. 2014; Akhlaq, Sheikh, and Pagliari 2015) and comparison with HICs(Afrizal et al. 2019).

2. The authors tried to conceptualize readiness in EHR; nevertheless, the introduction section lacks the necessary motivation and reflection explaining readiness in relation to low- and middle-income setting.

Response: We have revised the introduction to describe the different forms or constructs of EHR readiness in the context of LMICs in the last part of paragraph 3, Line 83-89 which now reads:

“Different forms of readiness have been described in literature with some overlap in meaning. Examples include core (need/motivational) readiness, technological (infrastructural) readiness, societal readiness, engagement readiness and learning (IT skills) readiness. The relative importance of each of these forms of readiness varies between organizational contexts. For example, poor IT infrastructure and lack of IT skills are often a barrier for EHR implementation in LMICs making technological readiness relatively more important for LMICs.”

In paragraph 4 (line 90-106), we discuss how Weiner’s conceptualization of organizational readiness relates to the constructs in literature on EHR readiness in LMICs.

3. Applying the Paré model in Uganda is important to expand knowledge. However, the authors did not provide a strong rationale for adopting it (page 6; line number 114-117). Instead, the authors dedicated time explaining the Weiner’s theory of organisational readiness for change. What was the motivation for selecting the Paré model? What makes the Paré model relevant to apply within Uganda?

Response: We have added in the last paragraph of the introduction the motivation for the Parémodel. It now reads:

“In this study we aimed to determine which factors within the model by Paré et al. underlie perceived organizational readiness to implement an EHR in oncology in Low and Middle Income Countries (LMICs). We chose the Paré model because it measures readiness at organizational level compared to, for example, tools by Khoja et al. which measure readiness at the level of antecedent constructs. Moreover, the Paré model was developed and validated within the same context of cancer care as the Uganda Cancer Institute, where this study was conducted”

4. The authors used “factors” in the title. Then, they used “variables” in the abstract (page 2; line number 24). Later, they returned to the term “factors” in the introduction (page 6; line number 114). Please, can the authors be consistent in the use of terminology?

Response: Thank you for this suggestion. We have replaced “factors” with “variables” throughout the manuscript.

5. The authors indicated that the questionnaires were printed and distributed (page 7; line number 144). However, it is not clear how participants were approached. Kindly provide details.

Response: We have re-written the methods section to give more clarity on participants’ selection and questionnaire distribution (line 177 – 189). Now it reads as follows:

“Eligible participants were all UCI staff who are directly involved in patient care or directly use the EHR. There are approximately 250 of these staff. Printed questionnaires were used for data collection, and they were distributed by the first author who is one of the clinical staff at the UCI. Additionally, staff who were not on site during the survey period (September to October 2018), e.g. due to study leave or other travels, were excluded since paper questionnaire were used. Online questionnaires were discouraged by the research ethics committee because there was no way to stamp them and assure the participants of official approval.

To minimize ordering effect (31), we made five versions of the questionnaire containing exactly the same items but with their order randomly shuffled using the free online list randomizer (32). The questionnaires were in English. One hundred and seventy-five questionnaires were distributed and 146 were returned (83.4% response rate).”

6. Authors indicated that not all UCI staff were on-site during the survey (page 8; line number 162-164). Please can the authors explain why they were unable to use an alternative approach such as an online-based or email-based survey to recruit the other staff?

Response: We appreciate the suggestion of using online questionnaires and we have considered this. However, as described above, only paper questionnaires were approved by the research ethics committee. In the revised methods section we include this explanation (line 184-185).

7. Regarding the inclusion criteria, staff years of EHR experience was not considered. Why?

Response:In Table 2 several aspects of digital literacy are described such as frequency of computer usage, self-assessed computer proficiency, experience of using an EHR, and received training on EHR.We collected EHR experience as a binary variable since EHR implementation in Uganda is limited and only begun in recent years, and we believe that frequency of computer usage is equally or more informative.

8. What was the relationship between the researcher who distributed the questionnaires and the participants prior to the study commencement? How did such a relationship influence the feedback? What strategies were integrated into the study to prevent/reduce potential bias from such researcher-participant relationship?

Response: The author who distributed the questionnaires is a member of the clinical team (paragraph 3, line 180) with no administrative authority. We believe there was no undue influence from this relationship that could bias. In addition, research ethics were followed to avoid undue influence as described in the informed consent form (Supplementary Information S5).

9. “…an influential and respected person to champion the implementation process” (Page 28; line number 314-315). Can the authors comment on and discuss with other international readiness literature the role of champions?

Response: Thank you for this comment. We have added an explanation of the role of champions and cited relevant literature in our discussion (line 327-330) as follows:

“Champions are important for success of EHR implementation because, as early adopters with positive attitude and enthusiasm towards the impending change, they help in communicating the expected benefits to their peers and encourage them to adopt the change (Gui et al. 2020; Shea and Belden 2015; Luz et al. 2019; Makam et al. 2014).”

Reviewer #2: 

1. The referred Fig. 1 is missing from the documents.

Response: We are sorry that the reviewer was unable to find the figure. It was provided as a separate file (not part of the manuscript), as per the style instructions of the journal. It is captioned “Figure 1 – Model diagram”.

2. The tables can be re-organized to make them reader friendly.

3. Also, a number of them can be combined to reduce the number of tables

Response: Thank you for these comments. We have merged Table 5 into Table 3 and displayed it in landscape to make it more readable.

4. Other minor comments are as tracked in the word version of the manuscript attached

Response: Thank you for the edit suggestions and comments. We have addressed them in the revised manuscript and they are highlighted in the version with “Track changes”. For example, we have rephrased the sentence about considerations for successful EHR implementation (Abstract (line 41-42) and in Results (line 280-283)), and we have rewritten the 2nd and 3rd paragraphs of the Introduction to address the reviewer’s comments on workflow changes that come with EHR implementation, and structural readiness.

---

## [Decision Letter · Decision Letter 1]

25 May 2020

PONE-D-20-07679R1

Assessment of organizational readiness to implement an electronic health record system in a low-resource settings cancer hospital: structural equation modeling of survey data to identify relevant variables

PLOS ONE

Dear Dr. Kabukye,

The reviewers are satisfied that you have addressed their previous comments, but one of them has recommended simplifying the second part of your title to read "a cross-sectional survey study" rather than '"structural equation modeling of survey data to identify relevant variables".

We look forward to receiving your revised manuscript.

Kind regards,

Tim Luckett

Academic Editor

PLOS ONE

Reviewers' comments:

Reviewer's Responses to Questions

**Comments to the Author**

1. If the authors have adequately addressed your comments raised in a previous round of review and you feel that this manuscript is now acceptable for publication, you may indicate that here to bypass the “Comments to the Author” section, enter your conflict of interest statement in the “Confidential to Editor” section, and submit your "Accept" recommendation.

Reviewer #1: All comments have been addressed

Reviewer #2: All comments have been addressed

2. Is the manuscript technically sound, and do the data support the conclusions?

Reviewer #1: Yes

Reviewer #2: Yes

3. Has the statistical analysis been performed appropriately and rigorously? 

Reviewer #1: Yes

Reviewer #2: Yes

4. Have the authors made all data underlying the findings in their manuscript fully available?

Reviewer #1: Yes

Reviewer #2: Yes

5. Is the manuscript presented in an intelligible fashion and written in standard English?

Reviewer #1: Yes

Reviewer #2: Yes

6. Review Comments to the Author

Reviewer #1: The authors have addressed all comments. But I think the title could benefit from a reword. It is lengthy. Kindly replace "structural equation modeling of survey data to identify relevant variables" with "a cross-sectional survey"

Reviewer #2: The authors have taken time to address all the issues I raised in my previous review to my satisfaction.

7. PLOS authors have the option to publish the peer review history of their article (what does this mean?). If published, this will include your full peer review and any attached files.

Reviewer #1: Yes: Andrew Donkor

Reviewer #2: Yes: Olutosin A. Awolude

---

## [Author Response · Author response to Decision Letter 1]

26 May 2020

Review Comments to the Author

Reviewer #1

The authors have addressed all comments. But I think the title could benefit from a reword. It is lengthy. Kindly replace "structural equation modeling of survey data to identify relevant variables" with "a cross-sectional survey"

Response: Thank you for this suggestion. We have revised the title, and now it reads: “Assessment of organizational readiness to implement an electronic health record system in a low-resource settings cancer hospital: a cross-sectional survey”

---

## [Editor Report · Decision Letter 2]

2 Jun 2020

Assessment of organizational readiness to implement an electronic health record system in a low-resource settings cancer hospital: a cross-sectional survey

PONE-D-20-07679R2

Dear Dr. Kabukye,

We are pleased to inform you that your manuscript has been judged scientifically suitable for publication and will be formally accepted for publication once it complies with all outstanding technical requirements.

With kind regards,

Tim Luckett

Academic Editor

PLOS ONE
---

## [Editor Report · Acceptance letter]

5 Jun 2020

PONE-D-20-07679R2 

Assessment of organizational readiness to implement an electronic health record system in a low-resource settings cancer hospital: a cross-sectional survey 

Dear Dr. Kabukye:

I'm pleased to inform you that your manuscript has been deemed suitable for publication in PLOS ONE. Congratulations! Your manuscript is now with our production department. 

Kind regards, 

on behalf of

Dr. Tim Luckett 

Academic Editor

PLOS ONE